# Bi-modality medical images synthesis by a bi-directional discrete process matching method

## Abstract

Recently, medical image synthesis gains more and more popularity, along with the rapid development of generative models. Medical image synthesis aims to generate an unacquired image modality, often from other observed data modalities. Synthesized images can be used for clinical diagnostic assistance, data augmentation for model training and validation or image quality improving. In the meanwhile, the flow-based models are among the successful generative models for the ability of generating realistic and high-quality synthetic images. However, most flow-based models require to calculate flow ordinary different equation (ODE) evolution steps in synthesis process, for which the performances are significantly limited by heavy computation time due to a large number of time iterations. In this paper, we propose a novel flow-based model, namely bi-directional Discrete Process Matching (Bi-DPM) to accomplish the bi-modality image synthesis tasks. Different to other flow matching based models, we propose to utilize both forward and backward ODE flows and enhance the consistency on the intermediate images over a few discrete time steps, resulting in a synthesis process maintaining high-quality generations for both modalities under the guidance of paired data. Our experiments on three datasets of MRI T1/T2 and CT/MRI demonstrate that Bi-DPM outperforms other state-of-the-art flow-based methods for bi-modality image synthesis, delivering higher image quality with accurate anatomical regions.

## 1 Introduction

Medical imaging plays a pivotal role in clinical diagnosis, treatment planning, and monitoring of various health conditions. Various imaging modalities such as, Computed Tomography (CT), Magnetic Resonance Imaging (MRI), and Positron Emission Tomography (PET), are widely used in clinical workflows, each of which can provide unique and distinct structural, functional, and metabolic information that enhances the overall scope for making accurate and reasonable clinical decisions. Even with huge benefits, some imaging modalities such as PET and CT, come with risks of radiation exposure. Moreover, the acquisition of multi-modal images are costly and time-consuming, which may also result in potential artifacts due to long time scanning. Hence, obtaining high quality multi-modality images remains a practical challenge in various clinical applications.

Inspired by the success of generative models for natural images, medical image synthesis provides an efficient solution through the transformation from one source image modality to a desired target one. Medical image synthesis can be used for data augmentation for model training Zhang et al. (2020) and validation Hu et al. (2023). In clinical applications it can also be used for MRI-only radiation therapy treatment planning and super-resolution Armanious et al. (2020); Dayarathna et al. (2023). Many novel generative neural network structures and algorithms have emerged to enhance performance in medical image synthesis, for capturing complex non-linear relationship between different image modalities and generating synthetic images of high quality. In early time, Generative Adversarial Networks (GANs) Goodfellow et al. (2014) are commonly used as the basic model and numerous GAN-related methods are proposed for medical synthesis and have remarkable performances Suganthi et al. (2021); Cao et al. (2020); Nie et al. (2018); Zhu et al. (2017).

Recently, the emergence of diffusion-based methods offer a different while effective tool for image generation and also promote the development on medical image synthesis Dorjsembe et al. (2022); Pan et al. (2023); Özbey et al. (2023); Müller-Franzes et al. (2023). From of point of view of image

synthesis, the generation process of classical diffusion models can be viewed as generating an image from a Gaussian variable Ho et al. (2020); Song & Ermon (2019). Thus it can not be directly used to find a transformation between two specific image styles. Consequently, some flow-based models with similar network structure are put forward, which can generate impressive images with specified style or modality, such as Conditional Flow Matching (CFM) Tong et al. (2023) and Rectified Flow (RF) Liu et al. (2022). Generally speaking, the image synthesis process can be described by a flow ODE:

$$\frac{d\boldsymbol{X}_t}{dt} = \boldsymbol{v}(\boldsymbol{X}_t, t), \ \ 0 \leq t \leq 1, \tag{1}$$

where $\boldsymbol{v}(\cdot, \cdot)$ represents the velocity field. The objective is to convert $\boldsymbol{X}_0$ from a source distribution $p(\boldsymbol{x})$ to $\boldsymbol{X}_1$ that follows the target distribution $q(\boldsymbol{z})$. To ensure the process $\boldsymbol{X}$ satisfies the condition that $\boldsymbol{X}_0 \sim p(\boldsymbol{x})$ and $\boldsymbol{X}_1 \sim q(\boldsymbol{x})$, both CFM and RF have elaborately designed specific transport paths. Precisely, in Tong et al. (2023) the author puts forward a uniform framework via using the mixture of conditional probability, which generates various formulations, such as the basic CFM (I-CFM), optimal transport CFM (OT-CFM), and variance preserving CFM (VP-CFM). On the other hand, Liu et al. (2022) directly utilizes the interpolation between $\boldsymbol{X}_0$ and $\boldsymbol{X}_1$ as the probability path, which makes the transport process straight and non-crossing. Furthermore, both methods are trained via flow matching Lipman et al. (2022), which uses a neural network $\boldsymbol{u}_\theta(\cdot, \cdot)$ to approximate a velocity field $\boldsymbol{v}(\cdot, \cdot)$ in the sense of some metric $d(\cdot, \cdot)$. Correspondingly, the parameterized velocity field $\hat{\boldsymbol{u}}_{\theta^*}(\cdot, \cdot)$ is obtained as follows:

$$\hat{\boldsymbol{u}}_{\theta^*}(\cdot, \cdot) = \arg\min_\theta \mathbb{E}_{t \sim \mathcal{U}([0,1])} \mathbb{E}_{\boldsymbol{X}_t}[d(\boldsymbol{u}_\theta(\boldsymbol{X}_t, t), \boldsymbol{v}(\boldsymbol{X}_t, t))]. \tag{2}$$

In both RF and CFM, the velocity field is pre-definded as the linear interpolation between the source and target distributions. However, for the medical image from different modality, the intermediate states resulting from the interpolation may tend to lack meaningful interpretations. And more importantly, some paired images are available in most cases and the objective is not only generate high-quality synthesis images but also preserve the paired information throughout the synthesis process. For instance, the same anatomical region of a patient is suppose to retain consistent tissue structure between the CT and MRI images. Thus the pair information may be crucial to be well utilized for the synthesis. On the other hand, in synthesis process, it is cumbersome to calculate the ODE flow along time from zero to one step by step, for which a small step-size takes a considerable amount of time while a large step-size might not be efficient for generating high quality images. Consequently, the choice of the step-size in flow-based methods like RF and CFM is crucial and requires careful consideration for different tasks as well.

In this paper, we propose a novel flow-based method, namely bi-directional Discrete Process Matching (Bi-DPM). Our approach ensures consistency between the intermediate steps of the forward and backward equations to learn the transformation between a source image modality and a target modality. Unlike recent mainstream flow-based models, Bi-DPM does not impose constraints on the transport paths. Instead, it focuses on matching intermediate states at pre-selected time steps from both the forward and backward directions of the flow ODE. We design a loss function that handles both fully paired and partially paired data, making our method applicable to a wide range of real world scenarios. We conduct numerical experiments on various medical image modality transfer tasks, and the results demonstrate that Bi-DPM generates high-quality synthesized images, outperforming other flow-matching methods in terms of FID, SSIM, and PSNR metrics. Additionally, Bi-DPM allows for a faster transfer process, as larger ODE step sizes can be used. Finally, clinical evaluations of the synthesized medical images by doctors highlight the potential for clinical application.

## 2 METHODOLOGY

### 2.1 BI-DIRECTIONAL DISCRETE PROCESS MATCHING

Suppose that $\{\boldsymbol{x}_i\} \sim p(\boldsymbol{x})$ and $\{\boldsymbol{z}_i\} \sim q(\boldsymbol{z})$ are two set of bi-modality image observations respectively. Let $\{\boldsymbol{X}_t\}_{0 \leq t \leq 1}$ be a random process defined on time interval $[0, 1]$. Then considering the flow ODE in Eq. 1 with the given initial condition $\boldsymbol{X}_0 = \boldsymbol{x}$ and the reverse process with initialization

$X_1 = z$, we have

$$\begin{cases} \dfrac{d\boldsymbol{X}_t}{dt} = \boldsymbol{v}(\boldsymbol{X}_t, t), \;\; 0 \le t \le 1, \\ \boldsymbol{X}_0 = \boldsymbol{x}, \end{cases} \quad \begin{cases} \dfrac{d\boldsymbol{X}_t}{dt} = -\boldsymbol{v}(\boldsymbol{X}_t, t), \;\; 0 \le t \le 1, \\ \boldsymbol{X}_1 = \boldsymbol{z}. \end{cases} \tag{3}$$

Then it is obvious that when the velocity is known, we can obtain $\boldsymbol{X}_1 \sim q$ from $\boldsymbol{X}_0 \sim p$ via the ODE from time $t = 0$ to $t = 1$ and vise versa. More generally, for $\forall t \in [0, 1]$ we have that

$$\boldsymbol{X}_t = \boldsymbol{X}_0 + \int_0^t \boldsymbol{v}(\boldsymbol{X}_s, s | \boldsymbol{X}_0 = \boldsymbol{x}) ds = \boldsymbol{X}_1 - \int_t^1 \boldsymbol{v}(\boldsymbol{X}_s, s | \boldsymbol{X}_1 = \boldsymbol{z}) ds, \tag{4}$$

where $\boldsymbol{v}(\boldsymbol{X}_s, s | \boldsymbol{X}_0)$ and $\boldsymbol{v}(\boldsymbol{X}_s, s | \boldsymbol{X}_1)$ are both equal to $\boldsymbol{v}(\boldsymbol{X}_s, s)$, connecting $\boldsymbol{x}$ and $\boldsymbol{z}$. Figure 1 displays the overall process of Bi-DPM, whose main idea is to choose a sequence of time point $0 = t_0 < t_1 < \cdots < t_N = 1$ and request the value of ODE Eq. 3 coincides with each other on these time points. Precisely, suppose $\boldsymbol{u}_\theta(\cdot, \cdot)$ represents our neural network with parameters $\theta$, and we call the process defined in Eq. 3 the *forward process* and the *backward process* with regard to velocity field $\boldsymbol{u}_\theta(\cdot, \cdot)$ which is denoted by $\boldsymbol{X}^f$ and $\boldsymbol{X}^b$ respectively:

$$\boldsymbol{X}_t^f = \boldsymbol{X}_0 + \int_0^t \boldsymbol{u}_\theta(\boldsymbol{X}_s^f, s | \boldsymbol{X}_0 = \boldsymbol{x}) ds, \quad \boldsymbol{X}_t^b = \boldsymbol{X}_1 - \int_t^1 \boldsymbol{u}_\theta(\boldsymbol{X}_s^b, s | \boldsymbol{X}_1 = \boldsymbol{z}) ds$$

Then for each discrete time point $t_n$ we can use a one-step numerical ODE solver to estimate $\boldsymbol{X}_{t_n}$ from $\boldsymbol{X}_{t_{n-1}}$ in forward iteration and opposite for the backward process, which is defined as follows:

$$\begin{aligned} \boldsymbol{X}_{t_n}^f &= \boldsymbol{X}_{t_{n-1}}^f + \boldsymbol{u}_\theta(\boldsymbol{X}_{t_{n-1}}, t_{n-1})(t_n - t_{n-1}), \\ \boldsymbol{X}_{t_{n-1}}^b &= \boldsymbol{X}_{t_n}^b + \boldsymbol{u}_\theta(\boldsymbol{X}_{t_n}, t_n)(t_{n-1} - t_n). \end{aligned} \tag{5}$$

Here we use Euler formula for solving the ODE. Then we can use a metric $d(\cdot, \cdot)$ to measures the distance between $\boldsymbol{X}_{t_n}^f$ and $\boldsymbol{X}_{t_n}^b$ for $\forall n \in \{0, 1, \cdots, N\}$. Hence, we propose our training objective function as follows:

$$\mathcal{L}(\theta) = \sum_{n=0}^{N} w_n d(\boldsymbol{X}_{t_n}^f, \boldsymbol{X}_{t_n}^b), \tag{6}$$

where $w_n$ is the weight at time $t_n$. With different type of training data, we can choose different metric $d(\cdot, \cdot)$ to match the characteristic properly. Precisely, in our experiments, we consider both cases of totally paired datasets and partially paired datasets. For paired data, we use Learned Perceptual Image Patch Similarity Zhang et al. (2018)(LPIPS) as the metric $d(\cdot, \cdot)$ while for unpaired data, we take Maximum Mean Discrepancy Smola et al. (2006)(MMD) to measure the distance between them Dziugaite et al. (2015); Sutherland et al. (2016); Li et al. (2017). Precisely, suppose $\{(\boldsymbol{x}_i^p, \boldsymbol{z}_i^p)\}$ are paired data and $\{\boldsymbol{x}_m^u\} \sim p(\boldsymbol{x})$ are $\{\boldsymbol{z}_n^u\} \sim q(\boldsymbol{z})$ are unpaired data. Then the training loss for paired data and unpaired ones are given as

$$\begin{aligned} \mathcal{L}^p(\theta) &= \sum_i \sum_{n=0}^{N} \text{LPIPS}(\boldsymbol{x}_{i,t_n}^f, \boldsymbol{z}_{i,t_n}^b), \\ &= \sum_i \sum_{n=0}^{N} \frac{1}{H_l W_l} \sum_{h,w}^{H_l, W_l} \| w_l \odot \big[ \phi_l(\boldsymbol{x}_{i,t_n}^f)_{h,w} - \phi_l(\boldsymbol{z}_{i,t_n}^b)_{h,w} \big] \|_2^2. \end{aligned} \tag{7}$$

$$\begin{aligned} \mathcal{L}^u(\theta) &= \sum_{p,q} \sum_{n=0}^{N} \text{MMD}(\boldsymbol{x}_{p,t_n}^f, \boldsymbol{z}_{q,t_n}^b), \\ &= \sum_{n=0}^{N} \Big[ \frac{1}{m^2} \sum_{p,p'} k(\boldsymbol{x}_{p,t_n}^f, \boldsymbol{x}_{p',t_n}^f) + \frac{1}{n^2} \sum_{q,q'} k(\boldsymbol{z}_{q,t_n}^b, \boldsymbol{z}_{q',t_n}^b) \\ &\qquad - \frac{2}{mn} \sum_{p,q} k(\boldsymbol{x}_{p,t_n}^f, \boldsymbol{z}_{z,t_n}^b) \Big]. \end{aligned} \tag{8}$$

where the $\mathcal{L}^p$ and $\mathcal{L}^u$ are paired loss and unpaired loss respectively and $\theta$ are the trainable parameters of the velocity field model. The $\boldsymbol{x}_{i,t_n}^f$ represents the intermediate state of sample $\boldsymbol{x}_i$ at time $t_n$ in

the forward process while $z_{i,t_n}^b$ are the corresponding state in the backward process. In (7), $\phi_l$ represents the $l$-th layer of a pretrained VGG net Simonyan & Zisserman (2014) and $H_l, W_l$ are the height and width of the corresponding feature. In (8), the $k(\cdot, \cdot)$ is a fixed kernel function. Therefore, our empirical training loss is defined as

$$\mathcal{L}(\theta) = \mathcal{L}^p(\theta) + \lambda_u \mathcal{L}^u(\theta), \qquad (9)$$

where $\lambda_u$ is a hyperparameter that controls the weight of MMD between unpaired data. Especially, for totally paired dataset, we only use LPIPS as loss function and $\lambda_u$ is equal to 0 correspondingly.

On the other hand, after obtaining a well-trained velocity field $\boldsymbol{u}_{\theta^*}(\cdot, \cdot)$, we can synthesis from $\boldsymbol{X}_0(\boldsymbol{X}_1)$ to $\boldsymbol{X}_1(\boldsymbol{X}_0)$ along the forward (backward) ODE along the direction $t_0 \leftrightarrows t_1 \leftrightarrows \cdots \leftrightarrows t_N$ and the corresponding $\boldsymbol{X}_1^f$ ($\boldsymbol{X}_0^b$) can be regarded as the final synthesis results. The algorithms for training and synthesis process are illustrated in Algorithm 1 and Algorithm 2.

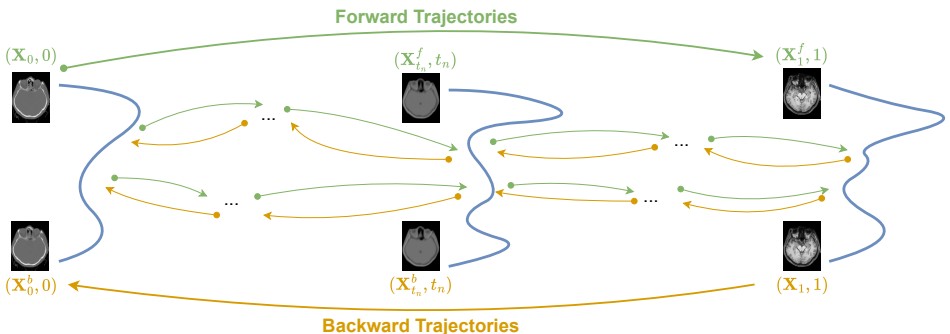

Figure 1: The overall pipeline of Bi-DPM.

---

**Algorithm 1:** Training of Bi-DPM

**Input:** time steps $\{t_0, t_1, \cdots, t_N\}$ with $t_0 = 0$ and $t_N = 1$, initial velocity model $\boldsymbol{u}_\theta(\cdot, \cdot)$, weight parameter $\{w_0, w_1, \cdots, w_N\}$, learning rate $\eta$, a metric $d(\cdot, \cdot)$.
**Data:** dataset $\mathcal{D}_1, \mathcal{D}_2$.

1 **repeat**
2      Sample $\boldsymbol{x} \sim \mathcal{D}_1$ and $\boldsymbol{z} \sim \mathcal{D}_2$;
3      Initialize $\boldsymbol{X}_0^f \leftarrow \boldsymbol{x}$ and $\boldsymbol{X}_1^b \leftarrow \boldsymbol{z}$ ;
4      **for** $n = 1, \cdots, N$ **do**
5          $\boldsymbol{X}_{t_n}^f \leftarrow \boldsymbol{X}_{t_{n-1}}^f + \boldsymbol{u}_\theta(\boldsymbol{X}_{t_{n-1}}^f, t_{n-1})(t_n - t_{n-1})$ ;
6          $\boldsymbol{X}_{t_{n-1}}^b \leftarrow \boldsymbol{X}_{t_n}^b + \boldsymbol{u}_\theta(\boldsymbol{X}_{t_n}^b, t_n)(t_{n-1} - t_n)$ ;
7      **end**
8      $\mathcal{L}(\theta) \leftarrow \sum_{n=0}^N w_n d(\boldsymbol{X}_{t_n}^f, \boldsymbol{X}_{t_n}^b)$ ;
9      $\theta \leftarrow \theta - \eta \nabla_\theta \mathcal{L}(\theta)$ ;
10 **until** *convergence*;

---

**Algorithm 2:** Synthesis on both direction via Bi-DPM

**Input:** well-trained velocity model $\boldsymbol{u}_{\theta^*}$, time steps $\{t_0, t_1, \cdots, t_N\}$ with $t_0 = 0$ and $t_N = 1$.
**Data:** initial sample $\boldsymbol{x} \sim \mathcal{D}_1$ and $\boldsymbol{z} \sim \mathcal{D}_2$

1 **for** $n = 1$ **to** $N$ **do**
2      $\boldsymbol{x} \leftarrow \boldsymbol{x} + \boldsymbol{u}_{\theta^*}(\boldsymbol{x}, t_{n-1})(t_n - t_{n-1})$;
3      $\boldsymbol{z} \leftarrow \boldsymbol{z} + \boldsymbol{u}_{\theta^*}(\boldsymbol{z}, t_n)(t_{n-1} - t_n)$;
4 **end**
     **Output:** $\boldsymbol{z}$ and $\boldsymbol{x}$

## 2.2 COMPARISONS TO OTHER METHODS

Flow-based methods, such as Rectified Flow (RF) or Conditional Flow Matching (CFM), emphasize aligning the entire of transport path. Specifically, both RF and CFM aim to minimize the following loss function with respect to a given true velocity field $v(\cdot, \cdot)$:

$$\mathcal{L}_{\text{continuous flow}}(\theta) = \mathbb{E}_{t \sim \mathcal{U}([0,1])} \mathbb{E}_{\boldsymbol{X}_t} \| u_\theta(\boldsymbol{X}_t, t) - v(\boldsymbol{X}_t, t) \|_2,$$

In RF, the velocity field is defined as $v(\boldsymbol{X}_t, t) = \boldsymbol{X}_1 - \boldsymbol{X}_0$, with the constraint $\boldsymbol{X}_t = (1-t)\boldsymbol{X}_0 + t\boldsymbol{X}_1$. And in CFM, the velocity field is defined as $v(\boldsymbol{X}_t, t) = \frac{\sigma'_t(z)}{\sigma_t(z)}(\boldsymbol{X}_t - \mu_t(z)) + \mu'_t(z)$, with the constraint $\boldsymbol{X}_t \sim \mathcal{N}(\mu_t(z), \sigma_t(z))$, where the variables $z$, $\mu_t(z)$ and $\sigma_t(z)$ are set differently, with each configuration leading to a distinct version of CFM.

In contrast, our Bi-DPM is designed to match the intermediate states at specific time points, which introduces more flexibility into the model and eliminates the need of a predefined velocity field. Furthermore, for each time points, Eq. 4 indicates the relationship:

$$\boldsymbol{X}_1 - \boldsymbol{X}_0 = \int_0^1 v(\boldsymbol{X}_s, s) ds$$

which can be regarded as a generalized version of the constraint $\boldsymbol{X}_0 - \boldsymbol{X}_1 = v(\boldsymbol{X}_t, t)$ used in RF.

**Remark 1** *Define $\Delta t = \max_{n=1,\cdots,N} \|t_n - t_{n-1}\|_1$ and suppose $u_\theta(\cdot, \cdot)$ is a solution to (6) with loss zero. Then if $u_\theta(\boldsymbol{X}_0, t_0) = u_\theta(\boldsymbol{X}_1, t_N)$ and $\Delta t \to 0$, it obtains that $u_\theta(\boldsymbol{X}_0, t_0) = \boldsymbol{X}_1 - \boldsymbol{X}_0$.*

**Proof** *For $\forall n \in \{1, \cdots, N\}$, following Eq. 5 and taking Taylor's expansion for each step, it obtains that*

$$\boldsymbol{X}_n^f = \boldsymbol{X}_0 + t_n u_\theta(\boldsymbol{X}_0, t_0) + o(\Delta t), \quad \boldsymbol{X}_n^b = \boldsymbol{X}_1 + (1 - t_n) u_\theta(\boldsymbol{X}_1, t_1) + o(\Delta t).$$

*Since $u_\theta(\cdot, \cdot)$ is a solution to (6) with loss zero, one gets that $\boldsymbol{X}_n^f = \boldsymbol{X}_n^b$ and correspondingly,*

$$\boldsymbol{X}_1 - \boldsymbol{X}_0 = u_\theta(\boldsymbol{X}_1, t_1) + o(\Delta t) = u_\theta(\boldsymbol{X}_0, t_0) + o(\Delta t),$$

*which leads to the conclusion in Remark 1 as $\Delta t \to 0$.*

According to Remark 1, the ground truth velocity field defined in RF is a specific solution to our problem. However, the objective of our model allows for solutions where the directions at $t = 0$ and $t = 1$ are both equal to $\boldsymbol{X}_1 - \boldsymbol{X}_0$, without imposing restrictions on the intermediate path during the transformation process. Furthermore, since our Bi-DPM focuses on points matching rather than relying on a predefined velocity field, it can fully leverage the paired relationship through the metrics such as LPIPS or $L_2$ distance in Eq. 9. In contrast, methods like RF and CFM struggle to effectively utilize the guidance provided by paired data.

As illustrated in Figure 2, we present a comparison using a toy example. In this setting, we aim to approximate the **nonlinear** transformation between two set of 8 Gaussians with different shape. Additionally, we assign part of paired relationships between the two sets. The star points in $\boldsymbol{X}_0$ and $\boldsymbol{X}_1$ represent the means of each Gaussian, and the green lines in Input indicate the correspondences between them. Except for the paired star points, all the remaining points are unpaired. As shown in the right three figures, while all the methods can generate a transformation between the two datasets, only our Bi-DPM is able to preserve the relationships between the paired data and accurately learn the transformation across the entire distribution under the guidance of the paired points. By comparsion, the RF and CFM exhibit poor performance and tend to converge to a "simplified" solution.

This toy experiment illustrates that RF and CFM may perform poorly when the true transformation is nonlinear, as their predefined velocity fields are constrained to be linear. In contrast, our Bi-DPM does not need rely on a predefined velocity field, but instead leverages the relationships between the paired points directly, which provides more flexibility in approximating nonlinear transformation.

## 3 EXPERIMENTS

We start from visualized 2D toy examples in Figure 2 and Figure 3 to demonstrate the effectiveness of the proposed model, with detailed illustrations provided in Section 2.2. Then we mainly focus

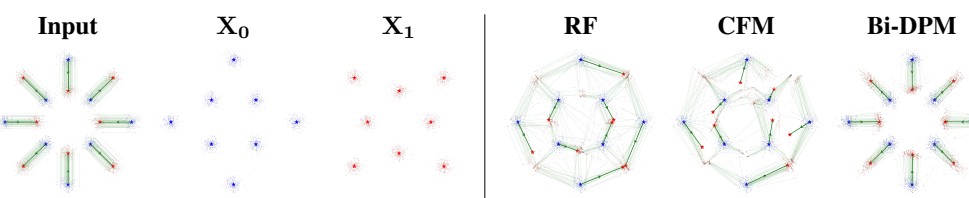

Figure 2: The performance of RF, CFM and Bi-DPM on the partially paired 8-Gaussian to 8-Gaussian toy example with the number of step is set to 10 for all methods.

on the synthesis task between different medical image modalities, including MRI T1-T2 and CT-MRI. And for image synthesis tasks, we evaluate our model on both totally paired and partially paired settings, providing some quantitative comparisons with several SOTA flow-based methods, along with image quality assessments. Additionally, we extend our model to 3D medical images synthesis, generating high-quality 3D images with visually superior results.

### 3.1 LOW DIMENSIONAL EXAMPLES

In addition to the example in Figure 2, we present two more cases involving two sets of 8 Gaussians with different paired data relationships. As shown in Figure 3, in both cases Bi-DPM can successfully approximates the relationships under the varying guidance from the paired data.

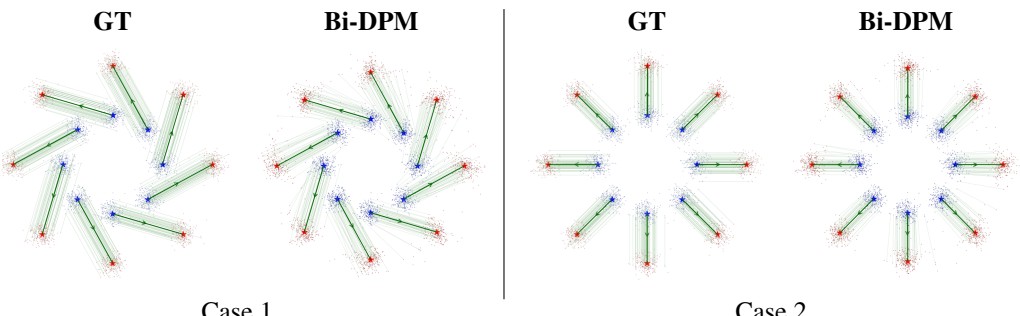

Figure 3: Toy Examples with different paired data relationships. In each case, the left figure represents the true relationship, and the right one illustrates the transformation learned by our Bi-DPM.

### 3.2 BI-MODALITY MEDICAL IMAGE SYNTHESIS

For medical image synthesis task, we perform a synthesis task between the medical image modalities, specially MRI T1/T2 and CT/MRI. The MRI T1/T2 dataset is sourced from BraTS 3D MRI images Baid et al. (2021); Menze et al. (2014) and the CT/MRI datasets are obtained from SynthRAD2023 images Thummerer et al. (2023). Since the original datasets contain three-dimensional images, we first extract 2D slices from each image to build our training and testing datsets. The MRI T1/T2 dataset comprises **1000** images pairs for training and **251** for testing. And for the CT/MRI task, we construct two datasets for different anatomical regions: the brain and the pelvis. Each dataset is split into **170** pairs for training and **10** for testing, among which we select 100 central slices for the brain and 50 central slices for the pelvis.

In training, all images are first resized to the resolution of $(192, 192)$ and then normalized to the range of $[-1, 1]$ Ho et al. (2020); Song & Ermon (2019). The step size for the $n-$step Bi-DPM is $1/n$, with the weights assigned as $w = 1$ for $t = 0, 1$ and $w = 0.5$ for all the intermediate states respectively. For all the experiments, we use UNet Ronneberger et al. (2015) structure parameterize the velocity field, as adopted in other flow-based methods Lipman et al. (2022); Liu et al. (2022); Tong et al. (2023). The optimizer for Bi-DPM is Adam Kingma & Ba (2014), with a constant learning rate of $10^{-4}$ in the training process. Besides, Exponential Moving Average Klinker (2011)(EMA) is used to update the flow-based models, and the two modality images are trained in pairs. Then we

compare our method against other SOTA transfer technieques such as CycleGAN Zhu et al. (2017), Conditional Flow Matching(CFM) Tong et al. (2023), Rectified Flow(RF) Liu et al. (2022). All results are evaluated with regard to Frechet Inception Distance Heusel et al. (2017)(FID, lower is better), Structure Similarity Index Measure Wang et al. (2004)(SSIM, higher is better), and Peak Signal-to-Noise Ratio Hore & Ziou (2010)(PSNR, higher is better). Because of space limitations, all the results related to CT/MRI Pelvis are provides in Supplementary Materials.

### 3.2.1 RESULTS WITH TOTALLY PAIRED DATA

The final comparison results on FID, SSIM and PSNR are summarized in Table 1. Due to space constraints, we display the synthesis results of BraTS MRI T1/T2 in Figure 4, with additional comparisons provided in Supplementary Materials. In Table 1, the Bi-DPM (1-step) and Bi-DPM (2-step) refer to time points set at $\{0, 1\}$ and $\{0, 0.5, 1.0\}$ respectively. For CFM methods, we evaluate various formulations proposed in Lipman et al. (2022), including the basic CFM (I-CFM), optimal transport CFM (OT-CFM) and variance-preserving CFM (VP-CFM). Additionally, for all the flow-based methods, we experimented with several different steps and selected the best-performing results for the synthesis process. As shown in Table 1, our method outperforms the other models across all three metrics (SSIM, FID and PSNR) on all the three tasks. Furthermore, for MRI T1/T2 task the 1-step Bi-DPM achieves the best results, while for both of CT/MRI experiments, the 2-step Bi-DPM yields optimal outcomes, which indicates that for images with complex structures, the inclusion of intermediate time points is both necessary and effective. Besides, as illustrated in Figure 4, the images generated by Bi-DPM preserve more details from the original input and are closer to the ground truth.

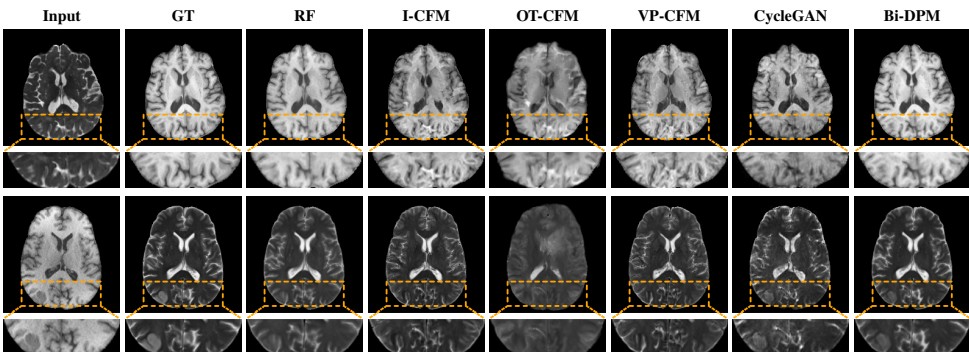

Figure 4: The synthetic images of **MRI T1/T2** dataset for different methods.

### 3.2.2 RESULTS WITH PARTIALLY PAIRED DATA

For the partially paired case, we use a combination of LPIPS and MMD as the loss function, as defined in (9), where LPIPS serves as the metric for paired data and MMD for unpaired data respectively. In our experiments, the weight $\lambda_u$ of MMD is fixed at either $0.2$ or $0.3$ during training. To further improve the training stability, each batch consists of an equal proportion of paired data and unpaired data, and the MMD is calculated between the unpaired data and all data in the same batch.

We conducted comparisons on the MRI T1/T2 and CT/MRI Brain datasets. Based on the experiments using fully paired data, we utilize the same training dataset but varied the proportion of paired data to 1%, 10% and 50%. The quantitative results of CT/MRI Brain dataset in terms of FID, SSIM and PSNR with respect to different ratio are presented in Figure 5. As shown, the quality of generated images improves as the ratio of paired data increases. Notably, our Bi-DPM achieves relatively high-quality performance with only 10% paired data, demonstrating that with even minimal partial guidance allows Bi-DPM to produce impressive results.

Additionally, Table 2 presents the quantitative results of Bi-DPM alongside other ODE-based methods. Based on the behaviors in Table 1, we only compare our method with the two best-performing methods, RF and I-CFM. The values in the brackets denote the corresponding results for the fully paired dataset, as shown in Table 1. Evidently, the performances of both RF and I-CFM are markedly

Table 1: Quantitative comparison on FID, SSIM and PSNR with 100% paired data. The **bold** data represent the best results and the underlined ones indicates the second best.

| | | RF | I-CFM | OT-CFM | VP-CFM | CycleGAN | Bi-DPM (1-step) | Bi-DPM (2-step) |
|---|---|---|---|---|---|---|---|---|
| MRI T1/T2 | T1 | SSIM ↑ | 0.841 ± 0.038 | 0.837 ± 0.040 | 0.729 ± 0.070 | 0.710 ± 0.042 | 0.652 ± 0.025 | **0.869** ± **0.038** | 0.862 ± 0.038 |
| | | FID ↓ | 49.004 | 85.069 | 168.269 | 63.826 | 80.771 | **40.611** | 57.800 |
| | | PSNR ↑ | 22.175 ± 2.385 | 21.843 ± 2.392 | 19.131 ± 2.294 | 18.648 ± 2.124 | 17.588 ± 1.716 | **23.117** ± **2.471** | 22.886 ± 2.449 |
| | T2 | SSIM ↑ | 0.833 ± 0.040 | 0.822 ± 0.040 | 0.666 ± 0.063 | 0.660 ± 0.044 | 0.633 ± 0.033 | **0.866** ± **0.041** | 0.857 ± 0.039 |
| | | FID ↓ | 37.825 | 41.780 | 163.686 | 50.477 | 68.749 | **32.366** | 38.960 |
| | | PSNR ↑ | 23.307 ± 1.839 | 22.743 ± 1.829 | 19.182 ± 1.857 | 18.843 ± 1.686 | 19.121 ± 1.257 | **24.845** ± **1.994** | 24.302 ± 1.944 |
| CT/MRI Brain | CT | SSIM ↑ | 0.828 ± 0.046 | 0.828 ± 0.046 | 0.672 ± 0.081 | 0.694 ± 0.058 | 0.650 ± 0.057 | 0.831 ± 0.046 | **0.832** ± **0.046** |
| | | FID ↓ | 62.425 | 70.691 | 136.472 | 81.583 | 108.584 | **33.426** | 34.557 |
| | | PSNR ↑ | 23.817 ± 1.856 | **24.122** ± **1.951** | 19.252 ± 1.956 | 21.366 ± 1.351 | 18.832 ± 0.985 | 23.656 ± 2.124 | 23.853 ± 2.080 |
| | MRI | SSIM ↑ | 0.678 ± 0.044 | 0.685 ± 0.046 | 0.467 ± 0.071 | 0.597 ± 0.047 | 0.445 ± 0.037 | 0.723 ± 0.047 | **0.726** ± **0.044** |
| | | FID ↓ | 56.972 | 85.528 | 147.784 | 72.462 | 116.487 | **29.991** | 31.452 |
| | | PSNR ↑ | 21.121 ± 1.174 | 21.131 ± 1.159 | 18.451 ± 1.521 | 19.650 ± 1.175 | 16.744 ± 0.927 | 21.140 ± 1.364 | **21.158** ± **1.340** |

inferior compared to the results with completely pairing, highlighting their strong dependence on the proportion of paired data. In contrast, our Bi-DPM incorporating the MMD loss for unpaired data, experiences only a slight decrease in performance compared to the fully paired case, demonstrating the robustness of Bi-DPM.

Table 2: Quantitative comparison on FID, SSIM and PSNR with 10% paired data.

| | | RF | | I-CFM | | Bi-DPM (1-step) | | Bi-DPM (2-step) | |
|---|---|---|---|---|---|---|---|---|---|
| MRI T1/T2 | T1 | SSIM ↑ | 0.587 ± 0.055 | (0.841 ±0.038) | 0.496 ± 0.053 | (0.837 ±0.040) | 0.840 ± 0.037 | (0.869 ±0.038) | **0.848** ± **0.038** | (0.862 ±0.038) |
| | | FID ↓ | 107.551 (49.004) | | 77.242 (85.069) | | **44.835** (40.611) | | 60.581 (57.800) | |
| | | PSNR ↑ | 16.520 ± 1.679 | (22.175 ±2.385) | 14.718 ± 1.404 | (21.843 ±2.392) | **21.862** ± **2.513** | (23.117 ±2.471) | 21.809 ± 2.224 | (22.886 ±2.449) |
| | T2 | SSIM ↑ | 0.557 ± 0.050 | (0.833 ±0.040) | 0.534 ± 0.041 | (0.822 ±0.040) | 0.828 ± 0.042 | (0.866 ±0.041) | **0.838** ± **0.040** | (0.857 ±0.039) |
| | | FID ↓ | 77.351 (37.825) | | 55.676 (41.780) | | **34.545** (32.366) | | 38.294 (38.960) | |
| | | PSNR ↑ | 17.054 ± 1.479 | (23.307 ±1.839) | 16.853 ± 1.258 | (22.743 ±1.829) | 22.796 ± 1.993 | (24.845 ±1.994) | **23.118** ± **1.962** | (24.302 ±1.944) |
| CT/MRI Brain | CT | SSIM ↑ | 0.735 ± 0.086 | (0.828 ±0.046) | 0.594 ± 0.099 | (0.828 ±0.046) | 0.803 ± 0.051 | (0.831 ±0.046) | **0.808** ± **0.051** | (0.832 ±0.046) |
| | | FID ↓ | 100.631 (62.425) | | 104.077 (70.691) | | **39.437** (33.426) | | 41.035 (34.557) | |
| | | PSNR ↑ | 20.332 ± 2.293 | (23.817 ±1.856) | 16.755 ± 2.180 | (24.122 ±1.951) | 22.770 ± 1.838 | (23.656 ±2.124) | **22.846** ± **1.786** | (23.853 ±2.080) |
| | MRI | SSIM ↑ | 0.545 ± 0.080 | (0.678 ±0.044) | 0.460 ± 0.082 | (0.685 ±0.046) | 0.678 ± 0.049 | (0.723 ±0.047) | **0.684** ± **0.053** | (0.726 ±0.044) |
| | | FID ↓ | 99.623 (56.972) | | 82.988 (85.528) | | **28.976** (29.991) | | 31.430 (31.452) | |
| | | PSNR ↑ | 18.087 ± 1.657 | (21.121 ±1.174) | 16.604 ± 1.304 | (21.131 ±1.159) | 20.685 ± 1.140 | (21.140 ±1.364) | **20.696** ± **1.241** | (21.158 ±1.340) |

### 3.2.3 SYNTHESIZED IMAGES QUALITY ASSESSMENT BY DOCTORS

To further evaluate the quality of the synthetic images, we invite three physicians from nationally high ranked local hospital for visual judgement, including an attending physician and two chief physicians. We set three levels of scores ranged from 0 to 2 for the realism of synthetic images, where score 0 indicates unrealistic and 2 indicates closed to real images. The test synthetic set consist of 5 MRT-T1, 5 MRI-T2, 5 Brain CT and 5 Brain MRI images. The results are presented in Table 3, with the scores representing the average ratings of 5 images for each modality. These

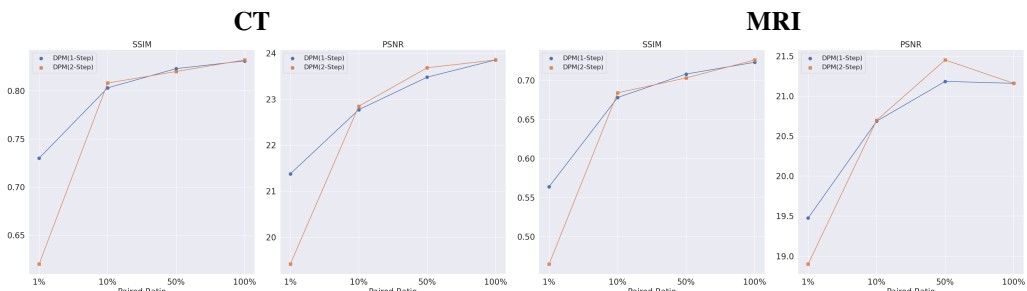

Figure 5: The quantitative results for the **CT/MRI Brain** dataset with various paired ratio. The left two columns show the results for synthetic CT images, while the right two columns correspond to synthetic MRI images. And for each modality, the evaluation indices include SSIM and PSNR.

results suggest that the majority of our synthetic images are judged as being close to realistic images. Specifically, only 3 images are rated as unrealistic (0 score) and 8 of them received a score of 1.

Table 3: The evaluations of three physicians (Average score of 5 images).

|  | MRI-T1 | MRI-T2 | CT | MRI | Average |
|---|---|---|---|---|---|
| Attending Physician | 1.8 | 2 | 1.6 | 1.2 | 1.65 |
| Chief Physician 1 | 1.8 | 2 | 2 | 1.8 | 1.9 |
| Chief Physician 2 | 1.2 | 2 | 2 | 2 | 1.8 |
| Average | 1.6 | 2 | 1.86 | 1.66 | 1.78 |

Additionally, a Turing test is conducted on 20 CT/MRI Brain images, consisting of 10 CT images and 10 MRI images. For each modality, there are 5 real images and 5 synthetic ones. The results of accuracy are shown in Table 4. As observed, only Chief physician 2 get accuracy above 50% while the other two physicians have an accuracy of only 40%, which indicates that our synthetic images are quite realistic and difficult to distinguish from real ones.

Table 4: The accuracy of Turing Test on Brain CT/MRI dataset.

|  | CT | MRI | Overall |
|---|---|---|---|
| Attending Physician | 20% | 60% | 40% |
| Chief Physician 1 | 30% | 50% | 40% |
| Chief Physician 2 | 50% | 60% | 55% |

Table 5: The quantitave comparison between Bi-DPM with the MRI-to-CT baseline.

|  | CT | | MRI | |
|---|---|---|---|---|
|  | SSIM | PSNR | SSIM | PSNR |
| Bi-DPM | **0.887** | **29.413** | 0.844 | 25.926 |
| Baseline | 0.871 | 29.307 | - | - |

### 3.2.4 3D IMAGES SYNTHESIS

We also apply our Bi-DPM to the task of 3D medical images synthesis. To deal with the 3D images, we slice each image along the transverse plane and convert it into a 2d task. Following the same setting as before, we resized the slices to (192,192) and scaled to the range of $[-1, 1]$ for training. During the transformation process, each slice is transferred from one modality to the other, and the slices are then reassembled in sequence. The experiments are conducted on the MRI-to-CT Brain task in SynthRAD2023 challenge, where we evaluate the performance of our Bi-DPM by comparing it against the baseline results from the competition leaderboard.

In the MRI-to-CT task, we follow the settings and make comparison to the baseline Chen et al. (2023), where the dataset is randomly split into 171 for training and 9 for testing. We calculate SSIM and PSNR for the generated 3D images, and the quantitative results are presented in Table 5. As shown, the synthetic CT images generated by our Bi-DPM achieve higher SSIM and PSNR values compared to the baseline. Moreover, with Bi-DPM we can also obtain the high-quality synthetic MRI images from the given CTs in the meanwhile, achieving SSIM of 0.844 and PSNR of 25.926. Additionally, comparisons between Bi-DPM-generated images and the ground truth for both CT and MRI are displayed in Figure 6.

**MRI-to-CT**            **CT-to-MRI**

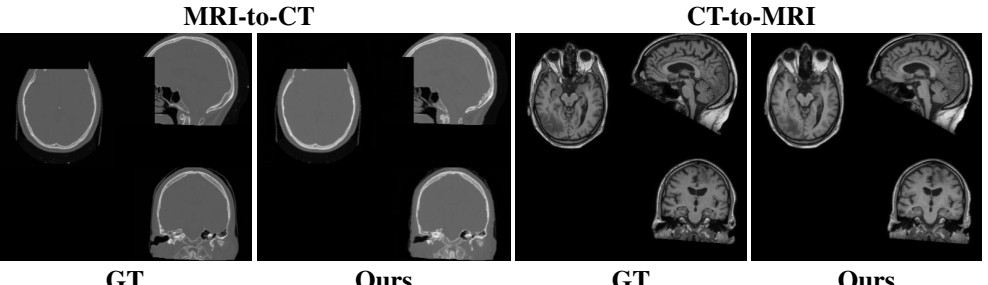

**GT**          **Ours**          **GT**          **Ours**

Figure 6: The synthetic 3D figures on axial, coronal and sagittal planes.

### 3.3 ITERATION STEPS OF ODE

In this part, we use totally paired CT/MRI Brain dataset to evaluate the influence of the number of ODE iteration steps on the synthesis process. The corresponding results for the other two datasets are displayed in Supplementary Materials. For all the other flow-based methods, we compare the synthesis results with 4 different ODE steps, including 1, 2, 5 and 10 steps. For simplicity, we treat CycleGAN as a one-step transformation method, and for our Bi-DPM, we calculate both one-step and two-step formulations. As shown in Figure 7, for VP-CFM, the evaluation indices improve as the number of ODE steps increases, which indicates that a higher number of steps is required to generate high-quality images in most cases. However, this comes at the cost of significantly increased computational demands. In contrast, for I-CFM, OT-CFM and RF, the results of 1-step perform best and the indices of multi-step remain relatively consistent or even degrade with more iteration steps, suggesting that the number of ODE steps needs careful tuning for each task to achieve optimal results, which complicates the synthetic process. However, despite of a slightly lower PSNR value compared to the best result of CFM on CT images with 1-step and 2-step generations, our Bi-DPM exhibits superior performances across both SSIM and PSNR indices, which makes Bi-DPM an effective model for generating high-quality images for both modalities.

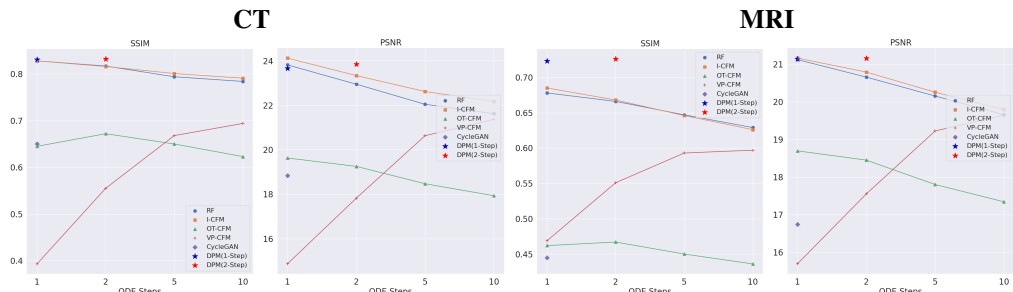

Figure 7: The quantitative comparison results on **CT/MRI Brain** dataset between different methods with various discrete ODE steps. The left two columns show the results for synthetic CT images, while the right two columns correspond to synthetic MRI images. For each modality, the evaluation metrics include SSIM and PSNR.

### 4 CONCLUSIONS

We propose a novel flow-based method, termed Bi-DPM for bi-modality images synthesis. Unlike other commonly used flow-based models, Bi-DPM accounts for both directions of the flow ODE and ensures the consistency in the intermediate states of the synthesis process. This approach effectively leverages the guidance from the paired data to generate high-quality synthetic images while preserving anatomical structure. Experiments on three independent datasets demonstrate that Bi-DPM outperforms other SOTA flow-based image transfer models in MRI T1/T2 and CT/MRI synthesis tasks.

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
