# A    Low Dimensional Examples

In this section we provide a detailed comparison of the toy example discussed in Section 2.2. In addition to the results of 10 steps, we also provide the outcomes of 2 steps and 5 steps for each method. As shown in Figure 8, both RF and CFM perform poorly with only 2 steps and 5 steps, whereas our Bi-DPM successfully learns the transformation between the two sets and preserves the relationships of the paired data in the meanwhile.

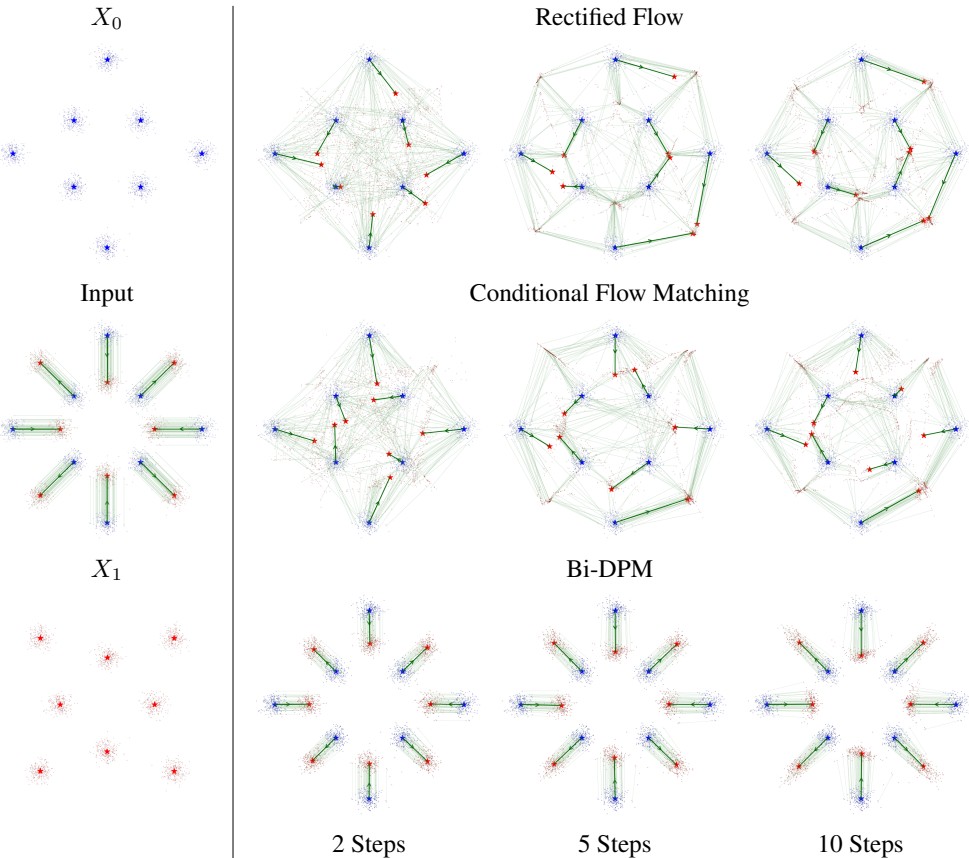

Figure 8: The performance of RF, CFM and Bi-DPM on the partially paired 8-Gaussian to 8-Gaussian toy example. For each methods, from left to right, the figure represents the results with ODE steps set to 2, 5, and 10.

Due to computation and memory constraints, here we test different number of steps on the toy examples using the datasets in Figure 8, and the L2 distance between the generated and true data is as follows:

Table 6: The $L_2$ error of different step sizes on the totally paired 8-Gaussian to 8-Gaussian toy example.

|  | 1-step | 2-step | 5-step | 10-step |
|---|---|---|---|---|
| $L_2$ error (forward/backward) | 0.015/0.015 | 0.009/0.008 | 0.011/0.012 | 0.013/0.019 |

As shown, 2-step achieves the best performance, while 1-step also performs comparably well compared to 5-step and 10-step. This partially justifies our choice of using only 1-step and 2-step in the image experiments. One intuition to use less steps is that the introduction of many intermediate steps may lead to unstable approximation, which may degrade the performance. A thorough stability analysis will be conduct in an ongoing work.

# B  Bi-Modality Medical Image Synthesis

## B.1  Results with totally paired data

We present more visualized comparisons on CT/MRI Brain and CT/MRI Pelvis datasets, which are presented in Figure 9 and Figure 10. Moreover, the quantitative results of the comparison to other baselines on CT/MRI Pelvis with totally paired data are displayed in Table 8.

Table 7: Quantitative comparison on FID, SSIM and PSNR with 100% paired data. The **bold** data represent the best results and the underlined ones indicates the second best.

| | | | RF | I-CFM | OT-CFM | VP-CFM | CycleGAN | SynDiff | Reg-GAN | Bi-DPM (1-step) | Bi-DPM (2-step) |
|---|---|---|---|---|---|---|---|---|---|---|---|
| MRI T1/T2 | T1 | SSIM ↑ | 0.841 ± 0.038 | 0.837 ± 0.040 | 0.729 ± 0.070 | 0.710 ± 0.042 | 0.652 ± 0.025 | 0.832 ± 0.041 | 0.809 ± 0.038 | **0.869** ± **0.038** | 0.862 ± 0.038 |
| | | FID ↓ | 49.004 | 85.069 | 168.269 | 63.826 | 80.771 | 97.494 | 85.767 | **40.611** | 57.800 |
| | | PSNR ↑ | 22.175 ± 2.385 | 21.843 ± 2.392 | 19.131 ± 2.294 | 18.648 ± 2.124 | 17.588 ± 1.716 | 20.377 ± 2.491 | 20.676 ± 1.926 | **23.117** ± **2.471** | 22.886 ± 2.449 |
| | T2 | SSIM ↑ | 0.833 ± 0.040 | 0.822 ± 0.040 | 0.666 ± 0.063 | 0.660 ± 0.044 | 0.633 ± 0.033 | 0.823 ± 0.043 | 0.805 ± 0.039 | **0.866** ± **0.041** | 0.857 ± 0.039 |
| | | FID ↓ | 37.825 | 41.780 | 163.686 | 50.477 | 68.749 | 49.568 | 88.087 | **32.366** | 38.960 |
| | | PSNR ↑ | 23.307 ± 1.839 | 22.743 ± 1.829 | 19.182 ± 1.857 | 18.843 ± 1.686 | 19.121 ± 1.257 | 21.965 ± 1.945 | 21.589 ± 1.451 | **24.845** ± **1.994** | 24.302 ± 1.944 |
| CT/MRI Brain | CT | SSIM ↑ | 0.828 ± 0.046 | 0.828 ± 0.046 | 0.672 ± 0.081 | 0.694 ± 0.058 | 0.650 ± 0.057 | 0.796 ± 0.056 | 0.817 ± 0.045 | 0.831 ± 0.046 | **0.832** ± **0.046** |
| | | FID ↓ | 62.425 | 70.691 | 136.472 | 81.583 | 108.584 | 70.172 | 38.606 | **33.426** | 34.557 |
| | | PSNR ↑ | 23.817 ± 1.856 | **24.122** ± **1.951** | 19.252 ± 1.956 | 21.366 ± 1.351 | 18.832 ± 0.985 | 22.344 ± 1.966 | 23.148 ± 1.790 | 23.656 ± 2.124 | 23.853 ± 2.080 |
| | MRI | SSIM ↑ | 0.678 ± 0.044 | 0.685 ± 0.046 | 0.467 ± 0.071 | 0.597 ± 0.047 | 0.445 ± 0.037 | 0.503 ± 0.052 | 0.683 ± 0.050 | 0.723 ± 0.047 | **0.726** ± **0.044** |
| | | FID ↓ | 56.972 | 85.528 | 147.784 | 72.462 | 116.487 | 71.952 | 31.856 | **29.991** | 31.452 |
| | | PSNR ↑ | 21.121 ± 1.174 | 21.131 ± 1.159 | 18.451 ± 1.521 | 19.650 ± 1.175 | 16.744 ± 0.927 | 17.172 ± 0.713 | 20.436 ± 1.339 | 21.140 ± 1.364 | **21.158** ± **1.340** |

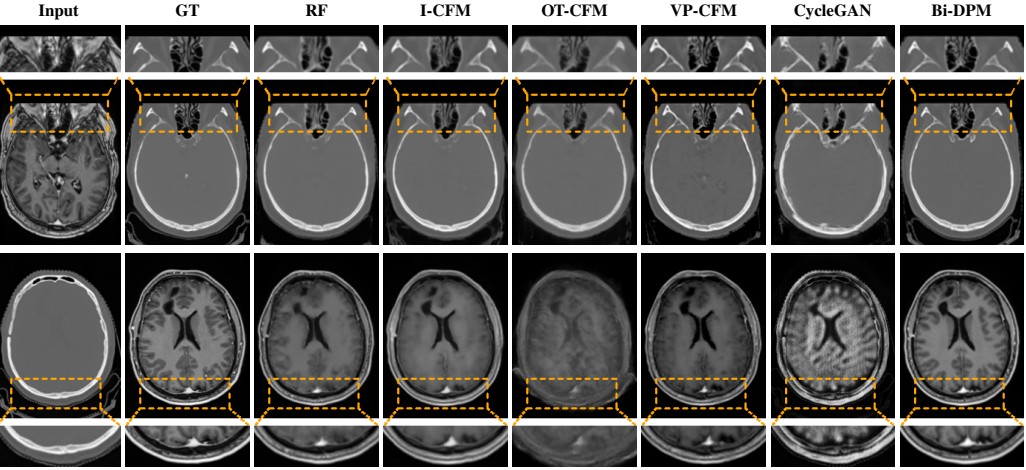

Figure 9: The synthetic images of **CT/MRI Brain** dataset for different methods.

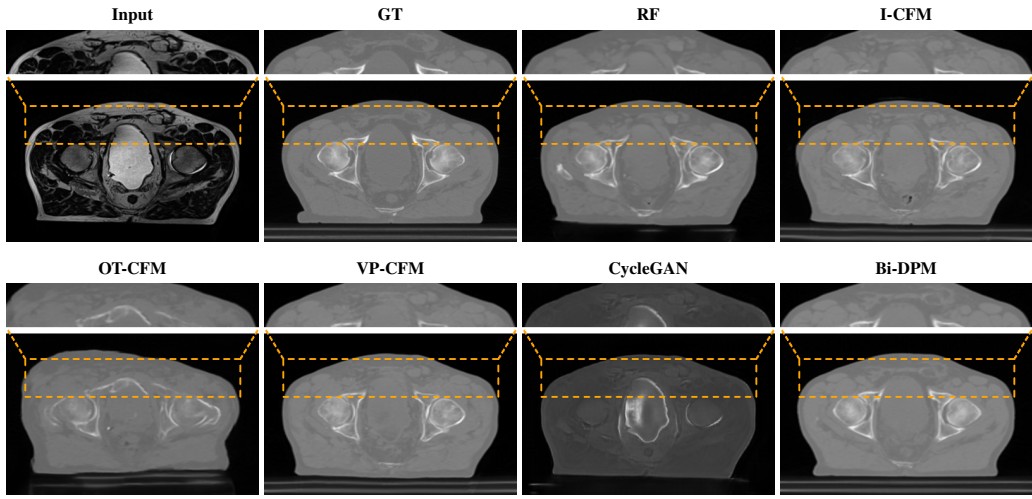

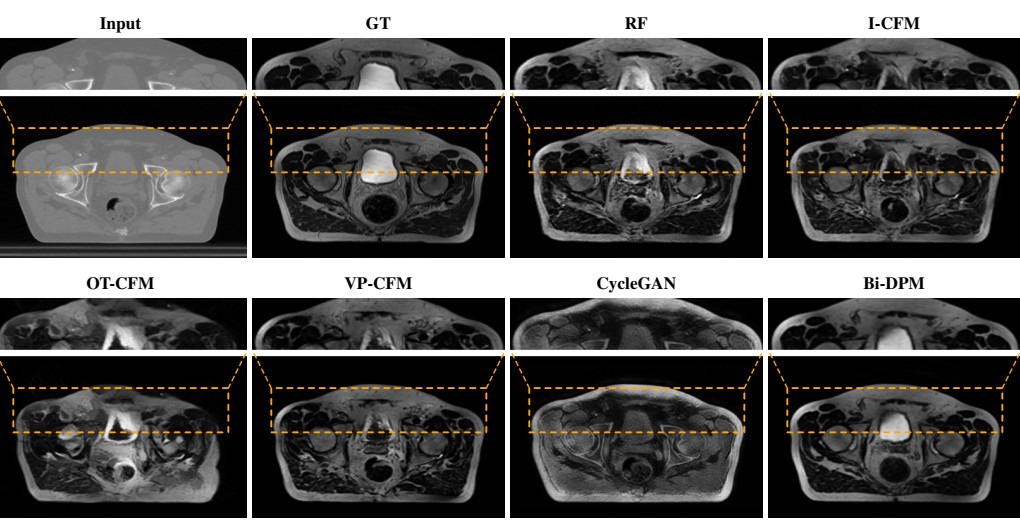

Figure 10: The synthetic images of **CT/MRI Pelvis** dataset for different methods.

Table 8: Quantitative comparison on FID, SSIM and PSNR with 100% paired data. The **bold** data represent the best results and the underlined ones indicates the second best.

| | | | RF | I-CFM | OT-CFM | VP-CFM | CycleGAN | Bi-DPM (1-step) | Bi-DPM (2-step) |
|---|---|---|---|---|---|---|---|---|---|
| CT/MRI Pelvis | CT | SSIM ↑ | **0.815** ± **0.046** | 0.810 ± 0.042 | 0.788 ± 0.048 | 0.710 ± 0.058 | 0.642 ± 0.054 | 0.806 ± 0.052 | 0.812 ± 0.049 |
| | | FID ↓ | 71.626 | 72.556 | 107.349 | 68.122 | 99.398 | **46.600** | 48.363 |
| | | PSNR ↑ | 23.591 ± 2.598 | 23.779 ± 2.279 | 22.104 ± 2.538 | 22.010 ± 1.932 | 19.672 ± 2.128 | 23.796 ± 2.313 | **24.033** ± **2.596** |
| | MRI | SSIM ↑ | 0.535 ± 0.052 | 0.532 ± 0.054 | 0.471 ± 0.081 | 0.415 ± 0.038 | 0.231 ± 0.042 | 0.571 ± 0.051 | **0.577** ± **0.052** |
| | | FID ↓ | 73.998 | 71.110 | 108.017 | 83.008 | 138.037 | 68.381 | **68.209** |
| | | PSNR ↑ | 17.892 ± 1.568 | 17.641 ± 1.497 | 16.090 ± 2.460 | 16.364 ± 1.512 | 14.428 ± 1.338 | 18.675 ± 1.687 | **18.930** ± **1.668** |

## B.2 RESULTS WITH PARTIALLY PAIRED DATA

The quantitative results of the comparison to other baselines on CT/MRI Pelvis with 0.1 ratio of paired data are displayed in Table 9. Besides, more results about the tendency of the quality evalution indices with regard to paired ratio are shown in Figure 11 and Figure 12.

Table 9: Quantitative comparison on FID, SSIM and PSNR with 10% paired data.

| | | | RF | | I-CFM | | Bi-DPM (1-step) | | Bi-DPM (2-step) | |
|---|---|---|---|---|---|---|---|---|---|---|
| CT/MRI Pelvis | CT | SSIM ↑ | 0.705 ± 0.038 | (0.815 ±0.046) | 0.684 ± 0.057 | (0.810 ±0.042) | 0.783 ± 0.053 | (0.806 ±0.052) | **0.785** ± **0.057** | (0.812 ±0.049) |
| | | FID ↓ | 147.101 | (71.626) | 102.730 | (72.556) | **52.517** | (46.600) | 54.528 | (48.363) |
| | | PSNR ↑ | 18.518 ± 1.692 | (23.591 ±2.598) | 19.381 ± 2.085 | (23.779 ±2.279) | 22.531 ± 2.306 | (23.796 ±2.313) | **22.847** ± **2.373** | (24.033 ±2.596) |
| | MRI | SSIM ↑ | 0.339 ± 0.078 | (0.535 ±0.052) | 0.255 ± 0.063 | (0.532 ±0.054) | 0.523 ± 0.048 | (0.571 ±0.051) | **0.532** ± **0.056** | (0.577 ±0.052) |
| | | FID ↓ | 138.121 | (73.998) | 166.533 | (71.110) | **76.436** | (68.381) | 77.927 | (68.209) |
| | | PSNR ↑ | 13.696 ± 2.662 | (17.892 ±1.568) | 12.062 ± 2.566 | (17.641 ±1.497) | 17.718 ± 1.559 | (18.675 ±1.687) | **17.917** ± **1.760** | (18.930 ±1.668) |

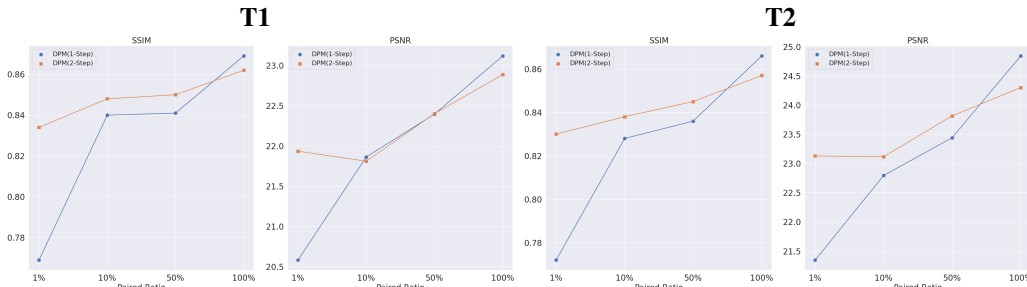

Figure 11: The quantitative comparison results on **MRI T1/T2** dataset between different methods with various discrete ODE steps. From the top to the bottom, the figures show the results of synthetic CT and synthetic MRI. The indices are SSIM, and PSNR from left to the right.

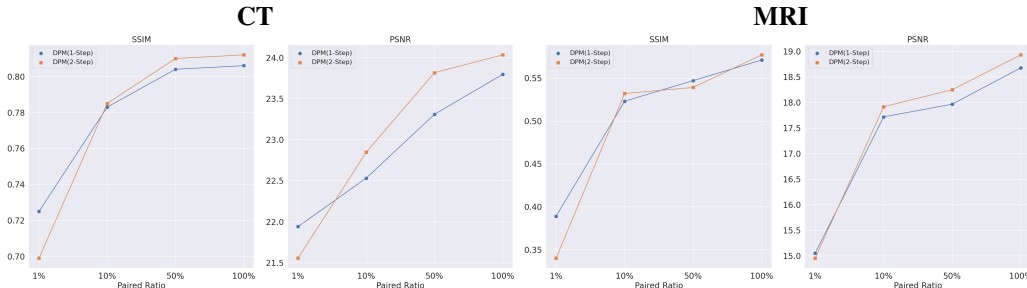

Figure 12: The quantitative comparison results on **CT/MRI Pelvis** dataset between different methods with various discrete ODE steps. From the top to the bottom, the figures show the results of synthetic CT and synthetic MRI. The indices are SSIM, and PSNR from left to the right.

## B.3   ITERATION STEPS OF ODE

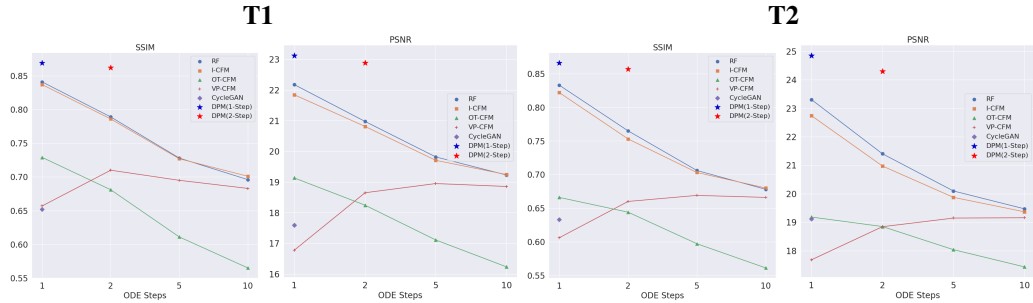

Figure 13: The quantitative comparison results on **MRI T1/T2** dataset between different methods with various discrete ODE steps.

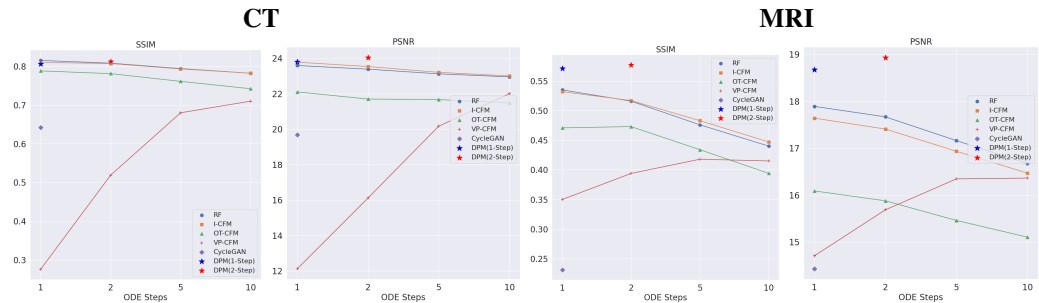

Figure 14: The quantitative comparison results on **CT/MRI Pelvis** dataset between different methods with various discrete ODE steps.

## C   TIME AND MEMORY COST

We also compare the memory cost in training process and the synthesis time cost for different methods. In training process, the batch size is set to 10. In testing, based on the MRI T1/T2 dataset, which contains $251 \times 2$ test images totally, we calculate the synthesis time cost of the whole dataset. For each method, we synthesize one image at a time and here are the results:

Table 10: Inference time cost comparison

|  | RF (RK45) | CFM (RK45) | CycleGAN | Reg-GAN | SynDiff | DPM (1-step) | DPM (2-step) |
|---|---|---|---|---|---|---|---|
| Time Cost | 834s | 834s | 31s | 10s | 394s | 20s | 38s |

Table 11: Training memory cost comparison

|  | RF (RK45) | CFM (RK45) | CycleGAN | Reg-GAN | SynDiff | DPM (1-step) | DPM (2-step) |
|---|---|---|---|---|---|---|---|
| Memory Cost | 26G | 36G | 22G | 12G | 56G | 48G | 63G |

## D   SEGMENTATION RESULTS

To assess the quality of the synthetic images, we evaluate their performance in a downstream segmentation task, representing a practical application of image synthesis. Using the MRI T1/T2 dataset from BraTS2021, we first train an nnUNet on real T1 and T2 training dataset. Subsequently, we test various combinations of synthetic and real data: baseline (T1 real and T2 real), T1 fake + T2 real and T1 real + T2 fake. The corresponding dice scores predicted by nnUNet are as follows:

Table 12: The segmentation results on MRI T1/T2 datasets.

|  | RF | CFM | CycleGAN | Reg-GAN | SynDiff | DPM |
|---|---|---|---|---|---|---|
| T1 Fake T2 Real | 0.814 | 0.812 | 0.758 | 0.774 | 0.781 | **0.816** |
| T2 Fake T1 Real | 0.690 | 0.662 | 0.519 | 0.642 | 0.619 | **0.716** |
| Baseline Both Real | 0.818 | | | | | |