# OpenReview forum: "Bi-modality medical images synthesis by a bi-directional discrete process matching method"
_ICLR.cc/2025/Conference — Submitted to ICLR 2025_

### Official Review · Reviewer_Vdxx · 2024-10-31

**Soundness:** 3
**Presentation:** 3
**Contribution:** 3
**Rating:** 6
**Confidence:** 4

**Summary:**

The paper proposes a novel bi-directional discrete process matching method to model the bi-modality image synthesis tasks. This method utilizes the both forward and backward ODE flow and enhances the consistency across each intermediate images. The method only requires a few discrete time steps for inference. The method can be used for variety of paired and unpaired datasets.

**Strengths:**

- The paper proposes novel bi-directional flow-based method for medical two modality synthesis, each can take advantage of paired information in two modality and keep consistent cross modalities.
- The paper is well-written and easy to follow.
- The paper considers variety of use cases including 3D image synthesis, pairs and unpaired datasets.

**Weaknesses:**

- For diffusion-based generative method, after modification, these models can be used for two-modality synthesis task.
- To keep each intermediate step matching on the flow trajectory from backward and forward direction, it might hurt the ability the model's generalization ability.
- Figures are too small to read.

**Questions:**

- Can the authors provide more comparison between your method with other diffusion based method, such as latent diffusion model?
- Can the authors provide some experiments on unseen data to test the generalization of your proposed method, such as EGD dataset?Chakrabarty, Satrajit, et al. "MRI-based classification of IDH mutation and 1p/19q codeletion status of gliomas using a 2.5 D hybrid multi-task convolutional neural network." Neuro-Oncology Advances 5.1 (2023): vdad023.
- Can the authors provide some comparisons between the effects on step size for MRI synthesis of proposed method, since there is only 1-step and 2-step size results?

---

> ### Author Response · Authors · 2024-11-25
>
> **Response to Weakness:**
>
> 1. Indeed there is some misunderstanding. We want to emphasize that, most diffusion model based methods require two or more networks to approximate the transformation in both directions separately. In contrast, our method DPM, achieves efficient bidirectional transfer using a single and simple network.
>
> 2. While it may slightly impact the model’s generalization ability, our approach still offers significant advantages. Utilizing a single network, our method accelerates both the training and synthesis processes, reduces model complexity, and still maintains robust generalization performance. This balance between efficiency and effectiveness highlights the strength of our model.
>
> **Response to Questions:**
>
> 1. We add the experiments of a diffusion-based model (2024), called SynDiff from paper Ozbey, Muzaffer, et al. “Unsupervised medical image translation with adversarial diffusion models.” IEEE Transactions on Medical Imaging 42. 12(2023): 3524-3539.  Here are the results and in () is the result of our model DPM:
>
> **SynDiff:** 2024
>
> **MRI T1/T2:**
>
> T1: SSIM: 0.823 (0.869); PSNR: 21.965 (23.117)
>
> T2: SSIM: 0.832 (0.866); PSNR: 20.377 (24.845)
>
> **CT MRI Brain:**
>
> CT: SSIM: 0.796 (0.832); PSNR: 22.344 (23.853)
>
> MRI: SSIM: 0.503 (0.726); PSNR: 17.172 (21.158)
>
> 2. The mentioned EGD dataset is not publicly available which can not be timely acquired, thus we are not able to provide a result on this dataset as requested.
>
> 3. Due to computation and memory constraints, we tested different number of steps on the low dimensional toy examples using the datasets show in Figure 2 instead of MRI figures to show the results of more step sizes, and the L2 distance between the generated and true data is as follows:
>
> **1-step**: 0.015/0.015; **2-step**: 0.009/0.008; **5-step**: 0.011/0.012; **10-step**: 0.013/0.019.
>
> As shown, 2-step achieves the best performance, while 1-step also performs comparably well compared to 5-step and 10-step. This partially justifies our choice of using only 1-step and 2-step in the image experiments. One intuition to use less steps is that the introduction of many intermediate steps may lead to unstable approximation, which may degrade the performance.

---

### Official Review · Reviewer_81s6 · 2024-11-03

**Soundness:** 3
**Presentation:** 3
**Contribution:** 4
**Rating:** 5
**Confidence:** 4

**Summary:**

This manuscript presents a novel flow-based method called Bi-directional Discrete Process Matching(Bi-DPM). The method utilizes both forward and backward ODE flows and enhance the consistency on the intermediate images to maintain high-quality generation under the guidance of paired data. Notably, it achieves significant improvements in PSNR, FID and SSIM, demonstrating the superior performance.

**Strengths:**

1.	Innovative Approach: This manuscript introduces an innovative flow-based model of medical image synthesis techniques to enhance the consistency on the intermediate images over discrete time steps in flow-based models, which helps maintaining pair information through synthesis process.
2.	Significant Empirical Improvements: The method substantially improves PSNR, FID and SSIM scores, demonstrating its effectiveness over existing methods.
3.	Detailed Methodological Framework: This manuscript presents a well-structured and comprehensive methodological framework, introducing the use of both forward and backward flow ODEs to preserve paired information.

**Weaknesses:**

1.	This manuscript lacks of sufficient description of the motivation and necessity of using both forward and backward ODE flows, I can’t see the necessity of this operation. The authors may add more detailed description on this.
2.	There lacks of implementation details. I would suggest the authors to add some description on their implementation.

**Questions:**

1.	The authors claim that diffusion models can not be directly used to perform image-to-image translation tasks, I wonder how do the authors draw this conclusion. I have read some researches doing this task using diffusion models, I recommend the authors to read these relevant references and add a comparison experiment with diffusion model. They are listed below:
[1]Graf, Robert, et al. "Denoising diffusion-based MRI to CT image translation enables automated spinal segmentation." European Radiology Experimental 7.1 (2023): 70.
[2]Ozbey, Muzaffer, et al. “Unsupervised medical image translation with adversarial diffusion models.” IEEE Transactions on Medical Imaging 42. 12(2023): 3524-3539.
[3]Kim, Jonghun, et al. “Adaptive Latent Diffusion Model for 3D Medical Image to Image Translation: Multi-modal Magnetic Resonance Imaging Study.” IEEE Winter Conference on Applications of Computer Vision 2023: 7604-7613.
2.	The only GAN used for comparison is CycleGAN, which is proposed in 2017. It seems out of date to me. I strongly suggest the authors to add comparisons with more advanced GAN models.
3.	The authors mentioned that the proposed Bi-DPM allows for a faster transfer process. I suggest the authors to add an experiment showing superior image synthesis speed to support their conclusion.
4.	I noticed that in the partially paired data results, 2-step Bi-DPM performs poorer than 1-step Bi-DPM when the paired ratio is low, can the authors add more detailed explanation on this phenomenon?

---

> ### Author Response · Authors · 2024-11-25
>
> **Response to Weakness:**
>
> 1. Ideally, establishing a one-to-one correspondence between the source and target datasets via a velocity field $v$ should ensure that the forward and backward states are perfectly aligned at every intermediate time step, based on the fundamental assumption of smoothness. This idea inspires us to incorporate the consistency of forward and backward flows at intermediate steps, thereby implicitly enforcing a smoothness constraint on the velocity field.
>
> 2. We are uncertain about the specific details that the reviewer is referring to. To address this, we have included more detailed comparisons with other methods and discussions on parameter selection. We would also be grateful if the reviewer could specify which parts require further clarification.
>
> **Response to Questions:**
>
> 1. Indeed there is some misunderstanding. We want to emphasize that, most diffusion model based methods require two or more networks to approximate the transformation in both directions separately. In contrast, our method DPM, achieves efficient bidirectional transfer using a single and simple network.
>
> 2. We include Reg-GAN as a new GAN-based method, which was published in 2021. There appears to be limited work for GAN in recent years and also draw comparison a diffusion based method. Also, we add the results of SynDiff, from the paper you mentioned in [2] Ozbey, Muzaffer, et al. “Unsupervised medical image translation with adversarial diffusion models.” IEEE Transactions on Medical Imaging 42. 12(2023): 3524-3539, as a diffusion-based method for comparison.
>
> **Reg-GAN:** 2021
>
> **MRI T1/T2:**
>
> T1: SSIM: 0.809 (0.869); PSNR: 20.676 (23.117)
>
> T2: SSIM: 0.805 (0.866); PSNR: 21.589 (24.845)
>
> **CT MRI Brain:**
>
> CT: SSIM: 0.817 (0.832); PSNR: 23.148 (23.853)
>
> MRI: SSIM: 0.683 (0.726); PSNR: 20.436 (21.158)
>
> **SynDiff:** 2024
>
> **MRI T1/T2:**
>
> T1: SSIM: 0.823 (0.869); PSNR: 21.965 (23.117)
>
> T2: SSIM: 0.832 (0.866); PSNR: 20.377 (24.845)
>
> **CT MRI Brain:**
>
> CT: SSIM: 0.796 (0.832); PSNR: 22.344 (23.853)
>
> MRI: SSIM: 0.503 (0.726); PSNR: 17.172 (21.158)
>
> 3. We provide quantitative comparisons of training and synthetic process between our DPM and other methods. The results are summarized as follows:
>
> **Training Time:** 20 batches/epoch and batch size 10. **Time per epoch:** RF: 7s, CFM: 10s, DPM (1-step): 13s , DPM (2-step): 17s.
>
> **Synthesis Time:** Using the MRI T1/T2 dataset, (251 * 2 test images totally), the total synthesis time for an image is as followed:
>
> DPM (1-step): 20s, DPM (2-step): 38s. Both RF and CFM: 834s (RK45 with adaptive step size is employed as proposed in the paper). Cycle-GAN: 31s; Reg-GAN: 10s; SynDiff: 394s.
>
> We note that our DPM achieves nearly the best performance while maintaining competitive efficiency. In training, DPM slightly increases time compared to RF and CFM but remains efficient. In testing, DPM's 1-step (20s) and 2-step (38s) are only slightly slower than Reg-GAN (10s) and Cycle-GAN (31s) respectively, but significantly faster than flow-based models like RF and CFM with RK45 formula and diffusion types approach (syndiff), while producing higher-quality results.
>
> 4. In our view, when the paired ratio is low, only a quite small portion of the data benefits from precise matching, while the majority relies on MMD, which may lead to less effective alignment across the entire dataset. In these cases, increasing the number of intermediate state may amplify the approximation errors, resulting in relatively poor performance. However, when the paired ratio is high, the stronger supervision from the paired data ensures better stability.

---

> ### Author Response · Authors · 2024-11-28
> **More experiments results and detailed explanations**
>
> We have added more experiments and detailed explanations of our model. We kindly encourage you to review our updated results.

---

### Official Review · Reviewer_4JLt · 2024-11-03

**Soundness:** 2
**Presentation:** 1
**Contribution:** 2
**Rating:** 3
**Confidence:** 3

**Summary:**

This paper presents Bi-Directional Discrete Process Matching (Bi-DPM) for bi-modality medical image synthesis. Bi-DPM employs a bi-directional process to align intermediate states across discrete time steps in both forward and backward directions with flow models. This method introduces a weighted "meeting point" loss, which includes terms for paired and unpaired data, allowing it to handle both types of data. By leveraging this bi-directional matching, Bi-DPM captures complex relationships between modalities, such as MRI T1/T2 and CT/MRI.

**Strengths:**

The method proposes a bi-directional recipe for the flow-matching models. Compared to methods like RF and CFM, the authors propose to constrain the consistency between intermediate states instead of restricting the velocity field to the difference. This allows a non-linear translation, as is shown in Fig. 2 and 3.

The 2D toy results shows convincing results on preserving the bi-direction relationship, even with few paired data.

**Weaknesses:**

The methodology may have an advantage over RF and CFM, but the authors have picked a bad application scenario. My major concern is the applicability of such methods in medical imaging. Unlike style translation in natural images, anatomy consistency is the utmost crucial factor in translating images between modalities. The MMD in (8) seems to be a very weak constraint for anatomical consistency in unpaired data. Is it even doable to translate medical images? Why can you recover T1/T2 properties of protons in the magnetic field given the HU in CT, there is as far as I know no theoretical ground to support this.

The results involved a physician rating, but the point here is not about letting them tell if the image is realistic. One can produce realistic images with many methods, like styleGAN, cycleGAN, etc., and many methods can do the same thing. The authors should have asked the question, “does the translated image tell the same information as the real images from the machine?”. The answer is probably no, because the resultant SSIM of CT-MRI is below 0.70. With such a low SSIM, most of the diagnostic-crucial information cannot be recovered. The algorithm is making random guesses in many areas. However, the accuracy of details are important for this task in clinical scenarios.

The experiments were not well-conducted, lacking ablation studies on hyper-parameters like the weight of the unpaired data MMD loss, which can be important for the final performances. The datasets are too small for a solid evaluation of the methods (described in sec. 3.2).

The figures of the paper are relatively hard to read, and the font size of the legends in Fig.5/7 is too small.

**Questions:**

Notation inconsistency: “X1 follows the target distribution q(z)” (line 064) “X1~q(x)” (line 065), should be “X1~q(z)” I suppose. Line 132, “to measures”.

The paper doesn’t consider some state-of-the-art unpaired/partially paired data translation methods (e.g. Schrödinger Bridge methods and some GAN-based methods other than CycleGAN, which is relatively out of date.).

In the training settings, the paper uses an equal proportion of the unpaired data and the paired data (line 366). But the training data consists of different proportions of the paired data (1%, 10%, 50%, and 100% illustrated in the experiments). The methods of augmenting the data to balance it into equal proportions must be specified.

---

> ### Author Response · Authors · 2024-11-25
>
> **Response to Weakness:**
>
> 1. Image translation for different modality has many clinical applications as certain modality is crucial while not available for the assistance of clinical diagnosis due to many reasons.  We agree that anatomy consistency is crucial for such application and the feasibility is partially validated by the image quality measure such as PSNR and SSIM to the real images. The true-false physical ratings is a first criterion as not all of style transfer approach from natural images such as styleGAN/cycleGan produce “real enough” medical images as reported in many existing literature and also shown in our figure. We also agree that the synthesize images should be also validated by clinical diagnosis downstream tasks. For this, we also performed downstream task such as lesion segmentation from synthesized images. The results show that the segmentation accuracy on synthesized one is comparable to real ones. One of our ongoing work is for calibrated PET images synthesis without CT scanning (for dose reduction) and the region of interests obtained by synthesis one has the same clinical behavior as the one from real ones. These downstream tasks can show the feasibility of medical image translation using AI approaches.
>
> 2. In practice, paired data are scarce, especially in medical image synthesis applications. While MMD may not be sufficiently strong to constrain two sets of unpaired images, we address this limitation by effectively leveraging partial paired data as guidance and MMD as an inline metric to better align the distributions. The numerical results with varies real MRI T1/T2, CT-MRI validate the capability of MMD measure with a small portion paired data as supervision.
>
> 3. For the 3D images settings, we preprocessing the data by slicing the 3D images into 2D slices for training and testing, resulting in a significantly large dataset of around 30k training samples. We believe this dataset size is substantial and adequately demonstrates the efficiency and effectiveness of our model in handling large datasets.
>
> **Response to Questions:**
> 1. We add the experiments of Reg-GAN (2021) and a diffusion-based model (2024), here are the results and in () is the result of our model DPM:
>
> **Reg-GAN**: 2021
>
> **MRI T1/T2**:
>
> T1: SSIM: 0.809 (0.869); PSNR: 20.676 (23.117)
>
> T2: SSIM: 0.805 (0.866); PSNR: 21.589 (24.845)
>
> **CT MRI Brain**:
>
> CT: SSIM: 0.817 (0.832); PSNR: 23.148 (23.853)
>
> MRI: SSIM: 0.683 (0.726); PSNR: 20.436 (21.158)
>
> **SynDiff**: 2024
>
> **MRI T1/T2**:
>
> T1: SSIM: 0.823 (0.869); PSNR: 21.965 (23.117)
>
> T2: SSIM: 0.832 (0.866); PSNR: 20.377 (24.845)
>
> **CT MRI Brain**:
>
> CT: SSIM: 0.796 (0.832); PSNR: 22.344 (23.853)
>
> MRI: SSIM: 0.503 (0.726); PSNR: 17.172 (21.158)
>
> We can see that our results still perform the best. Further comparisons on computation and memory cost and downstream segmentation task are also shown in the updated manuscript.
>
> 2. The purpose of using different proportions of paired data (1%, 10%, 50%, 100%) in training dataset is to demonstrate the robustness of our model. In all these settings, each training batch maintains an equal proportion of paired data and unpaired data, ensuring stable and balanced model training.

---

> ### Author Response · Authors · 2024-11-28
> **More experiments results and detailed explanations**
>
> We have added more experiments and detailed explanations of our model. We kindly encourage you to review our updated results.

---

### Official Review · Reviewer_vFFh · 2024-11-06

**Soundness:** 3
**Presentation:** 3
**Contribution:** 3
**Rating:** 6
**Confidence:** 3

**Summary:**

This paper introduces a flow-based generative model aimed at improving medical image synthesis across different modalities (e.g., MRI T1/T2 and CT/MRI). Unlike traditional flow-based models that focus on a unidirectional path, the proposed method leverages forward and backward Ordinary Differential Equations (ODEs) to ensure consistency across intermediate image states, enhancing the synthesis process's accuracy and quality.

**Strengths:**

x. Bi-DPM matches intermediate states at discrete time points between forward and backward ODEs, enhancing consistency and allowing high-quality image synthesis that preserves anatomical details.

x. Loss function flexibility: The model incorporates a loss function that can handle both fully paired and partially paired datasets using metrics like LPIPS for perceptual similarity and MMD for unpaired data.

x. Empirical validation: Experiments conducted on MRI T1/T2 and CT/MRI datasets show that Bi-DPM outperforms state-of-the-art flow-based models, such as Conditional Flow Matching (CFM) and Rectified Flow (RF), in terms of SSIM, PSNR, and FID scores.

x. Clinical relevance and 3D image synthesis: The paper includes evaluations by physicians, where Bi-DPM-generated images were deemed highly realistic, with a Turing test indicating the difficulty in distinguishing these images from real ones.

**Weaknesses:**

x. **Need more elaboration on mathematical derivations**: While I understand the overall purpose of the derivations, some of the deeper mathematical proofs and their implications, like those in Remark 1, could be more thoroughly explained or connected to the practical advantages of the model.

x. **Over-argument**: In Section 3.2.4, while the authors present the slicing approach as a straightforward extension of their 2D method, they do not sufficiently explain how this adapts to or addresses the complexities of 3D medical imaging. This simplification might give the impression that applying 2D techniques to 3D data is easier than it is in practice.

x. **Missing details on efficiency**: While Bi-DPM is described as computationally more efficient with larger step sizes, the paper does not provide detailed comparisons of training and inference times against other models. A computational resource analysis (e.g., time per training epoch, memory requirements) would be valuable for practical considerations.

**Questions:**

x. **ODE step size**: How sensitive is Bi-DPM to changes in the ODE step size? Did the authors test different step sizes systematically, and what were the observed trade-offs in terms of computational efficiency and image quality?

x. **Hyperparameter selection**: How were the weight parameters $w_n$ for the loss function chosen, and how do they impact the training stability and results?

x. **Physician feedback**: What qualitative feedback did the physicians provide during the image evaluation, and were there specific features or characteristics that made the synthetic images more or less convincing?

x. **Performance on unpaired data**: While Bi-DPM handles paired and partially paired data well, how does it perform on completely unpaired data? Is there a significant performance drop, and if so, what are the main challenges?

Minor: Segmentation may be considered a downstream task / additional metrics to evaluate image quality.

---

> ### Author Response · Authors · 2024-11-25
>
> **Response to Weakness**:
>
> 1. Thanks for the suggestion on more theoretical exploration. Indeed, a primary study show that the model is closed related to ODE-net without imposing explicit constraints on the velocity, which allows more flexible transport mapping from the source to the target, compared to other regularization-based flow-based models. A more through theoretical study and connections to other flow-based approaches is an ongoing work.
>
> 2. Due to high memory demand for 3D images in practice, a widely used approach to handle 3D medical image is to train and synthesis on the 2D transverse slices which evaluate on 3D images.  Our method takes this approach by training and synthesizing 2D slices and concatenate them to obtain 3D synthetic medical image.
>
> 3. The details on the training and synthetic process are summarized as follows and updated in the manuscript:
>
> **Training Time**:  20 batches/epoch and batch size 10.
> **Time per epoch**: RF: 7s, CFM: 10s, DPM (1-step): 13s , DPM (2-step): 17s.
>
> **Testing Time**: Using the MRI T1/T2 dataset, (251 * 2 test images totally), the total synthesis time for an image is as followed:
> DPM (1-step): 20s, DPM (2-step): 38s. Both RF and CFM: 834s (RK45 with adaptive step size is employed as proposed in the paper).
> Cycle-GAN: 31s; Reg-GAN: 10s; SynDiff: 394s.
>
> Our DPM achieves nearly the best performance while maintaining competitive efficiency. In training, DPM slightly increases time compared to RF and CFM but remains efficient. In testing, DPM's 1-step (20s) and 2-step (38s) are only slightly slower than Reg-GAN (10s) and Cycle-GAN (31s) respectively, but significantly faster than flow-based models like RF and CFM with RK45 formula and diffusion types approach (syndiff), while producing higher-quality results.
>
> Also, the training memory cost of our DPM is only a slightly higher than CFM but significantly lower than SynDiff, which demonstrates that within a minimal and acceptable increase of training memory, our DPM can achieves substantially better performance compared to other methods, highlighting its efficiency and effectiveness.
>
> **Response to Questions**:
>
> 1. We found that using few steps such as 1-step and 2-step is a good choice for both synthesis quality and computation cost. Due to computation and memory constraints, we tested different number of steps on the low dimensional toy examples using the datasets show in Figure 2, and the L2 distance between the generated and true data is as follows:
>
> 1-step: 0.015/0.015; 2-step: 0.009/0.008; 5-step: 0.011/0.012; 10-step: 0.013/0.019.
>
> 2-step achieves the best performance, while 1-step also performs comparably well compared to 5-step and 10-step. This partially justifies our choice of using only 1-step and 2-step in the image experiments. One intuition to use less steps is that the introduction of many intermediate steps may lead to unstable approximation, which may degrade the performance. A thorough stability analysis will be conduct in an ongoing work.
>
> 2. We set the weights at t=0 and t=1 as 1.0 and  the intermediate states as 0.5. One intuition of this setting is to force the model to match closely at t=0 and t=1, while less constraints on the generated intermediate steps. We also test on the toy example with the weight all equal to 1, the results are degraded with higher errors. We note that for different datasets, we fix this choice of parameter setting, thus the choice if rather stable with respect to different datasets.
>
> 3. In the current work, we perform a Turing test by asking the physicians to judge whether an image is real or synthesis one and score the image quality of generated images. We also perform downstream task to justify the usage of synthesis images by segmenting the lesions from synthesized images. The results show that the segmentation accuracy on synthesized one is comparable to on the real ones, which further validate the effectiveness of synthetic images.
>
> **Both Real**: 0.818519;
>
> **T1 Fake + T2 Real**: DPM: 0.816; RF: 0.814; CFM: 0.812; Reg-GAN: 0.774; CycleGAN: 0.758; Syndiff: 0.781
>
> **T1 Real + T2 Fake**: DPM: 0.716; RF: 0.690; CFM: 0.662; Reg-GAN: 0.642; CycleGAN: 0.519; Syndiff: 0.619
>
> As shown, our DPM achieves the best performance across all the settings. Notably, with real T2, DPM's segmentation results are closest to those one the true test dataset.
>
> 4. In practice, paired data are scarce, especially in medical image synthesis applications. Indeed, when applied to completely unpaired data, performance decreases as MMD may not be a strong measure to find a correspondence between different image modalities. However, it can still provide an inline measure of in-distribution and cross distribution distance, with an easy computation. Thus we propose to use a small portion of paired data as guidance and MMD is used as an inline metric to better align the distributions. The numerical results validate the capability of MMD measure with a small portion paired data.

---

> > ### Comment · Reviewer_vFFh · 2024-11-29
> >
> > Thank you. The authors clarified most points regarding the design choice and experiments.
> > The limitation I found is its application part in neuroimage synthesis which is often done in 3D in the latest work. 2D approaches cause intensity discontinuity between slides from other views (except the one used for training).
> >
> > Minor: The inference time for CycleGAN seems very long.

---

> ### Author Response · Authors · 2024-11-28
>
> We have further established a theoretical property of our method, demonstrating that if $v_\theta$ can make the objective function (loss function) equal to 0, we can get that $v_\theta(x_t, t) = X_1 - X_0$ for alll $t \in [0,1]$, consistent with the objective of Rectified Flow (RF).  However, different from RF, which relies on matching the velocity field, our approach directly match the image states. This enables a more precise alignment of the synthetic image details, enhancing the quality of the generated results.

---

> ### Author Response · Authors · 2024-11-30
>
> As illustrated in Figure 6 in our paper, we presented the results of slices from two additional views, showing only subtle discontinuity. This indicates that our method is capable of synthesizing high quality 3D images with consistent and reliable performance across different perspectives. Furthermore, with sufficient computational resources, our method can be directly extended to 3D application by replacing the original 2D modules with 3D modules in the network.

---

> > ### Comment · Reviewer_vFFh · 2024-11-30
> > **thank you**
> >
> > Thanks for pointing this out. Indeed the images are good.
> >
> > **To AC**: Overall, it's a good paper that could raise discussion at ICLR. I keep my rating as "borderline accept."

---

### Meta-Review · Area_Chair_vDSF · 2024-12-18

**Metareview:**

This paper presents to use bidirectional flow for medical image synthesis. This paper receives two slightly positive comments and two negative opinions. While the overall approaches look reasonable, there is a concern on the fidelity of the generated images. This has been raised by Reviewer 4JLt and I agreed with the reviewers. Meanwhile, the authors mentioned that "heavy computation time" is one of the limitation in existing methods, which makes me feel that they would try to address this in the paper. However, I cannot find any comparison on this. The authors may want to write the papers more carefully to make sure that the paper follows a order from the problem, existing limitation, proposed solution to solve the limitation and significance. In additional, the small figures also make it hard to tell how this paper improves the results. I believe that the authors do not have to visually compare with some many methods. Instead, they shall highlight how the proposed method (bidirectional flow) changes/improves the results.

**Additional Comments On Reviewer Discussion:**

NA

---

### Decision · Program_Chairs · 2025-01-22

Reject